# Hydroclimatic instability accelerated the socio-political decline of the Tang Dynasty in northern China

M. Kempf [1,2] ✉, M. L. C. Depaermentier [3], R. N. Spengler III [4], M. D. Frachetti [5], F. Chen [6,7], J. Luterbacher [8,9], E. Xoplaki [10] & U. Büntgen [2,11,12]

Extreme flooding and prolonged, intensifying droughts have played a critical role in the rise and collapse of preindustrial states and empires worldwide, triggering cascading impacts such as crop failure, famine, and migration that undermined socio-political stability and economic resilience. We present a multicomponent hydroclimatic vulnerability model for crop supply networks to estimate the contribution of climatic stressors as one of several factors contributing to the decline of the late Tang Dynasty in northern China between 800 and 907 CE. We demonstrate that recurrent flooding and prolonged droughts, combined with an unsustainable shift in crop production from drought-tolerant millet to less resilient wheat and rice, led to harvest failures and food shortages during the cooler and drier climatic conditions of the late 9th and early 10th centuries CE. Intensifying raiding from competing polities and climatic extremes further affected grain supplies for the late Tang's northern military frontier and partly contributed to the sudden decline of the dynasty. Our results emphasize the importance of multicomponent environmental response models to understand historical transformations and provide new aspects of China's socio-political development during medieval times.

The Yellow River Loop (YRL) region in northern China (Fig. 1), close to today's border with Mongolia, played a crucial role in the agricultural, political, and economic powers of ancient states in eastern Asia[1,2]. The YRL represented a politically contested transitional zone where competing interests of local communities and state authorities intersected in the struggle for resource control. The YRL was also strategically important due to its access to fertile land, salt, and pasture. The spatial distribution of cropland and climatically sensitive grasslands was, however, strongly affected by changes in seasonal hydroclimatic properties, precipitation amount and the extent of the East Asian Summer Monsoon (EASM)[3,4]. Access to resources further triggered political interventions to maintain food security during periods of hydrological and meteorological extremes, such as prolonged droughts and consecutive flooding events[5,6]. Characterized by

run-off values below normal, YRL hydrological droughts are amplified by anomalies in precipitation (P) and potential evapotranspiration (PET), leading to moisture deficits[7–10]. Multiannual events and agricultural droughts increase the vulnerability of crops and subsequent potential for famine caused by food shortages[7,11]. The oscillation and interplay of very wet conditions, enhanced flooding and prolonged droughts with sociopolitical internal and external stressors further accelerated the vulnerability of food security, particularly for premodern, peripheral subsistence economies in the northern YRL[12,13]. In addition, continuous impacts on the landscape caused by environmental shifts or human decision-making have reshaped agricultural affordances through time. Particularly, irrigation and manuring alter soil composition and fertility, leading to period-inherent agricultural demands that are rather situational than constant[14–16].

[1]Quaternary Geology, Department of Environmental Sciences, University of Basel, Basel, Switzerland. [2]Department of Geography, University of Cambridge, Cambridge, UK. [3]Faculty of History, Vilnius University, Vilnius, Lithuania. [4]Domestication and Anthropogenic Evolution Research Group, Max Planck Institute of Geoanthropology, Jena, Germany. [5]Department of Anthropology, Washington University in St Louis, St Louis, USA. [6]ALPHA Group, Institute of Tibetan Plateau Research, Chinese Academy of Sciences, Beijing, China. [7]Key Laboratory of Western China's Environmental Systems, College of Earth and Environmental Sciences, Lanzhou University, Lanzhou, China. [8]Department of Geography, Climatology, Climate Dynamics and Climate Change, Justus-Liebig University Giessen, Giessen, Germany. [9]Center for International Development and Environmental Research, Justus-Liebig University Giessen, Giessen, Germany. [10]CMCC Foundation – Euro-Mediterranean Centre on Climate Change, Bologna, Italy. [11]Department of Geography, Masaryk University, Brno, Czech Republic. [12]Czech-Globe, Global Change Research Institute CAS, Brno, Czech Republic. ✉e-mail: michael.kempf@unibas.ch

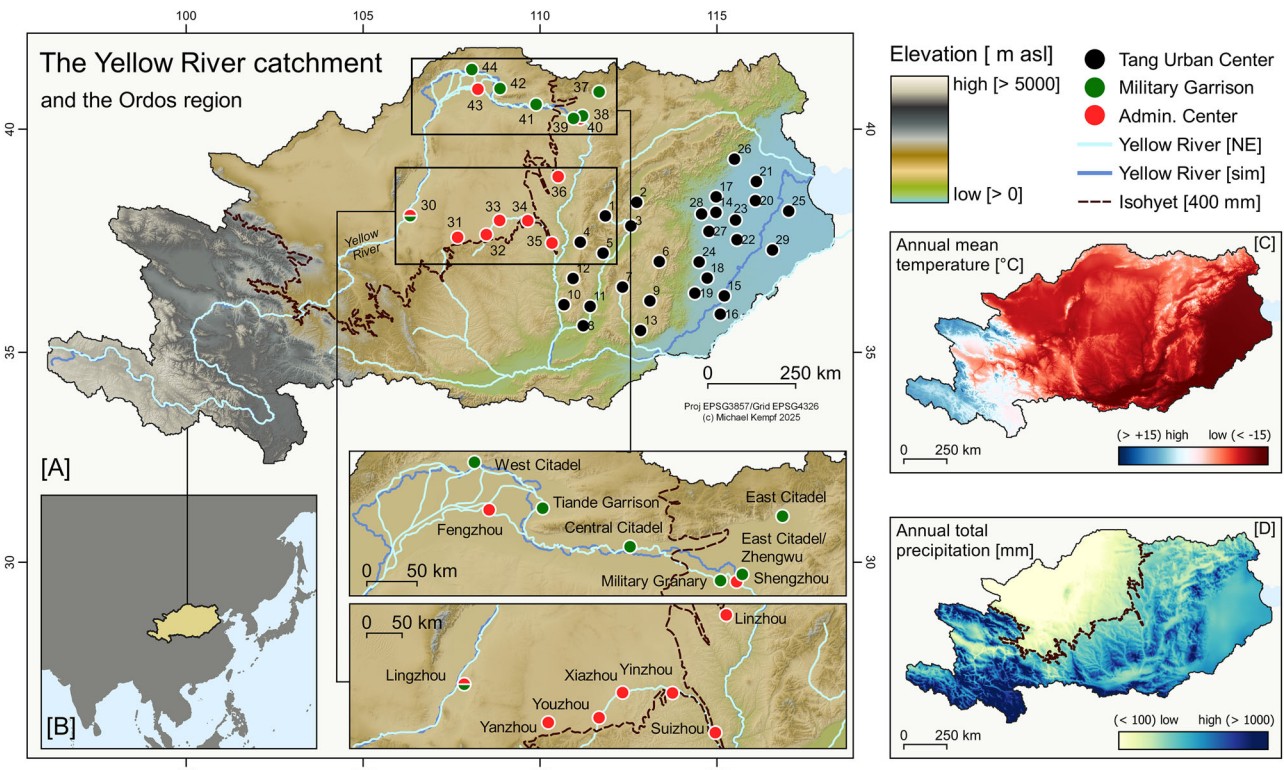

**Fig. 1 | Study area, site locations, place names, and environmental settings.** Topography (**A**), extent (**B**), annual average temperature (**C**) and precipitation (**D**)[91]. Yellow River simulation represents the simulated main stem based on topography; Yellow River [NE] is the Natural Earth river centerline dataset (naturalearthdata.com, last accessed 18th of March 2025); the dashed line represents the 400 mm/year isohyet, which is considered to determine the southern boundary of the farming-pastoral transition area and almost coincides with the EASM boundary in China (calculated from CHELSA monthly gridded data from 1980 to 2018 CE[91]). Tang towns subset from CHGIS[104,105]; last accessed 5th of April 2025 (see supplementary Table S1 for site names, location and dating. See Table S2 and Fig. S1 for the location of the prefectures).

Agricultural crop production is highly dependent on the climatic conditions during the growing season[17], with the selection of climate-adapted and resilient crop varieties playing a crucial role in preventing harvest failures. Despite the introduction of wheat into a previously millet-based agricultural system around 3000–2000 BCE[18,19], archaeobotanical and stable isotopic evidence from northern China indicates that millet remained the dominant crop in the human diet until at least the 5th century CE[20–22]. However, isotopic evidence from the Xinjiang region shows that wheat had already become an important dietary component by ~1800 BCE, while in the Gansu region, located in the upper reaches of the Yellow River, a mixed millet–wheat diet emerged from around 1700 BCE[22]. Further south, in the Zhengzhou area, the importance of wheat increased steadily from the Eastern Zhou period to the Han Dynasty (770 BCE to 220 CE)[23]. Along the middle and lower reaches of the Yellow River, substantial evidence for an increased dietary role of wheat does not appear until the 5th century CE[22,24–26].

During the Tang Dynasty in the north, wheat increasingly became an important part of the diet. It has been theorized that the later rise in prominence of wheat is tied to seasonal crop-rotation cycles. In the north, these cycles would have involved winter wheat raised on monsoonal precipitation and summer irrigated millets, legumes, and vegetable crops[27–29]. In the southern regions of the empire, rice was likely rotated with wheat, whereas wet paddies were drained for winter planting. It would not be until the Song Dynasty that a fast-growing and high-yielding rice, called by historians Chompas rice, would be introduced; multiple rice cycles on one plot of land would not have been possible before that time[30]. Crop-rotation cycles would have permitted greater yields near urban centers, therefore allowing for greater demographic density and fueling imperial expansion[17].

According to historical and archaeobotanical sources, the Tang Dynasty relied heavily on the staple crops millet (*Setaria italica* and *Panicum miliaceum*) and to a lesser extent wheat (*Triticum aestivum*) and rice (*Oryza sativa*) that shaped both diet and economy[31–33]. Particularly, the dominant millet was well adapted to harsh growing conditions during dry periods[34–36]. Wheat, however, became more and more prominent during the middle[37] and particularly the late Tang Dynasty[22,33,38]. The early and middle Tang Dynasty, spanning approximately from the 7th to the mid-8th centuries CE, benefited from overall warm temperatures and relatively humid conditions[6,39], which facilitated substantial agricultural expansion and enhanced resilience to political turmoil[40,41]. Such circumstances also resulted in substantial population growth[9,42]. Higher temperatures and more monsoonal precipitation likely contributed to a notable increase in grain yields[35,36]. However, towards the late Tang Dynasty, and particularly after 800 CE, relatively cooler and drier conditions affected the less drought-tolerant wheat and rice much more than millet, compromising integrated crop-rotations systems[40]. This climatic shift not only reduced agricultural productivity but also contributed to social and economic instability in the northern YRL, because poor harvests could no longer sustain the growing Tang population[42]. Major socio-political upheaval, migration towards the south and east, and constant political turmoil and warfare with groups along the northern borders with Mongolia further increased the vulnerability of the Tang's marginal frontier zone, leading to deployment of military troops and infrastructure development.

Consequently, a network of fortified centers was constructed in the 8th and 9th centuries CE to support the northwestern Tang frontier defense, hosting tens of thousands of soldiers[43–48]. Food supply relied on tax allocations in the form of grain, local crop production, and costly long-distance transport from the Tang's agricultural core production area in the eastern

and south-eastern part of the middle and lower Yellow River basins[49,50]. Located within the eastern bend of the YRL, Shengzhou was a transfer hub where grain from Taiyuan was stored in military granaries and transported to the administrative hubs and military garrisons. Recurring flood events between 812 and 825 CE, however, struck the region, damaging citadels and forcing the relocation of key military infrastructure[43,51].

Increased hydrological drought occurrences during the late Tang Dynasty in the 9th century CE[1,8,52] coupled with continuous conflict from the north-western plateau and internal rebellion[6,13,53] forced the local population to migrate towards the east and south, further disconnecting the food supply from the military troops at the northern frontier zones[41]. Economic and fiscal reforms, such as the introduction of commercial taxation, provided some relief to the late Tang Empire[54]. However, more pressure ultimately destabilized the government and contributed to the fragmentation and collapse of the dynasty in 907 CE[9,42].

Here, we argue for a more nuanced connection between proximal and ultimate causation, whereas a succession of internal conflicts and military campaigns ultimately ended the dynastic reign, but ecological stressors exacerbated the reduced political resilience[55]. The complex interplay of consecutive hydroclimatic extremes and cascading effects such as harvest failure, famines, and migration and economic vulnerabilities that accelerated the socio-political instability of the late Tang Dynasty, however, remains insufficiently explored. Understanding the contribution of famines and food shortages during periods of increased socio-political and environmental vulnerability is key to address the mechanisms and underlying dynamics behind the decline of the centralized Tang state power[13,56,57].

When debating the factors that lead to collapse or continuity of political entities, it is essential to differentiate between proximal and ultimate factors. Archaeological theorists have noted that external factors rarely provide the proximal cause of collapse, even if they are the ultimate cause[58,59]. While climatic events or military invasion may ultimately cause the destruction of an empire, that political system's inability to cope with the external factors is almost invariably due to internal dynamics. This is undoubtedly the case for the Tang Dynasty, whereas historians have already laid out the path of political turmoil that entailed the proximal causes. The Tang were plagued with internal conflict, including the An Lushan Rebellion, culminating in the sacking of the capital of Chang'an by the Tibetan Empire in 763[60]. The empire would never rebound after the Huang Chao Rebellion, and over the following two decades, a series of raids, internal coups, and social turmoil would lead to a final power change in 907 CE[61]. The success of these military rebellions was due, in large part, to a succession of external factors, including military campaigns with Central Asia (such as the Battle of the Talas River), raiding from peoples to the north, and environmental unpredictability.

Here, we suggest that a critical factor was that local and regional crop production and subsequent long-distance food supply networks became increasingly vulnerable to climatic conditions at the end of the 9th and the early 10th century CE based on the multicomponent model performance. As a consequence, supply networks adapted to increased environmental and climatic vulnerability, replacing insufficient regional production of grain with long-distance sourced foods from the Tang agricultural center. Transportation of grain, however, was an unsustainable method to cope with changes in dietary habits and crop production. We further argue that unsustainable land-use and shifts in crop production and dietary habits weakened the peripheral communities and destabilized the Tang's northern military frontier zones.

To explore the rapid decline of the Tang Dynasty, we evaluate a set of vulnerability models simulating food supply chains across the late Tang agricultural productivity zone during dry and humid extremes between 800 and 907 CE. We present (i) a multicomponent hydroclimatic vulnerability model for the YRL based on topographic, climatic, and hydrological variables; (ii) connectivity patterns of the late Tang administrative and military expansion into the northern YRL; and (iii) the reconstruction of late Tang crop production networks and the adaptation of military supply chains to past climate variability. We then provide a new interpretation of late Tang political decline caused by the interplay of climate variability, extremes and socio-economic instability. Our results stress the need for high-resolution multicomponent response models to understand past socio-cultural development in China. We further emphasize the high vulnerability of agricultural productivity to climatic variability including extremes in the northern YRL, and the importance of broad crop repertoires for sustainable land-use[62].

## Results and discussion

### Hydroclimatic vulnerability of the YRL

The multicomponent hydroclimatic vulnerability model presented here (Fig. 2) shows the areas with high risk potential during hydrological droughts combined with simulated flooding extent. The model distinguishes a clear zonation of high drought vulnerability in the central and northern part of the Yellow River catchment and the transition to a more moisture-balanced eastern part, affected by precipitation regimes of the EASM. Moisture availability is the major control factor of crop production in the northern part of the YRL. The present-day 400 mm/year isohyet indicates the approximate threshold for the transition between a monsoon-controlled climate and a stronger continental influence and is considered the southern boundary of the farming-pastoral transition area (Fig. 1)[1,63,64]. In the south-eastern part of the Yellow River, monsoonal rainfall mitigates the risk for droughts during periods of a stronger EASM, which accounts for just over 50% of the annual precipitation between June and August[65], however, it is also connected with a higher flood risk[66].

All urban centers of the late Tang Dynasty and their corresponding agricultural productivity areas are located in less drought-vulnerable areas in the lower basins of the Yellow River (Fig. 2A, B). In addition to the EASM precipitation patterns across the watershed, the flow accumulation of the branching lower Yellow River counteracts the strong negative gradient of the climatic water balance (CWB) and the high temperature in the lower basin (Figs. S2–S4). Historical sources confirm the development of extensive state-controlled river-based irrigation systems under the Tang Dynasty to cope with water shortages during dry periods[67–70].

In the northern part of the YRL, the modeled risk of crop failure increased strongly due to higher drought vulnerability, particularly following the Tang Dynasty's shift from drought-tolerant millet to less resilient wheat as the staple crop. The newly established administrative centers and the military fortifications of the late Tang period were located in areas with elevated drought risk, further compounded by high flooding potential. These sites were strategically positioned near the 400 mm/year isohyet (Fig. 1). The military sites show different spatial behavior and are located strategically in the northern YRL floodplain (Figs. S5 and S6). Crop production in this area is limited by the strongly negative CWB and the high drought vulnerability causing crop failures during years with decreased precipitation and increased PET. The multiannual monthly Standardized Precipitation Evapotranspiration Index (SPEI) drought index (Fig. S7) shows the high drought vulnerability of the newly established late Tang administrative and military centers in the northern YRL. In addition to the location in the northern bend of the YRL, the military fortifications are further prone to extensive flooding, which amplified the risk for crop failure and destruction of infrastructure during years with above-normal run-off. With limited space and increased vulnerability in agriculture, local crop production in the north was restricted to subsistence farming during periods with increased hydroclimatic extremes - above or below normal precipitation events.

### Late Tang Dynasty land-use strategies

During the late Tang Dynasty, crops were produced in the Tang agricultural heartland in the eastern part of the Yellow River catchment (Fig. 3). High soil quality together with abundance of running freshwater for centralized irrigation management and less drought vulnerability made the region highly productive (Figs. S9–S12). Across the entire prefecture, our land-use probability model shows high scores compared to the northern YRL (Figs. S13 and S14). The model includes soil properties and productivity scores from moisture availability, temperature, and topography.

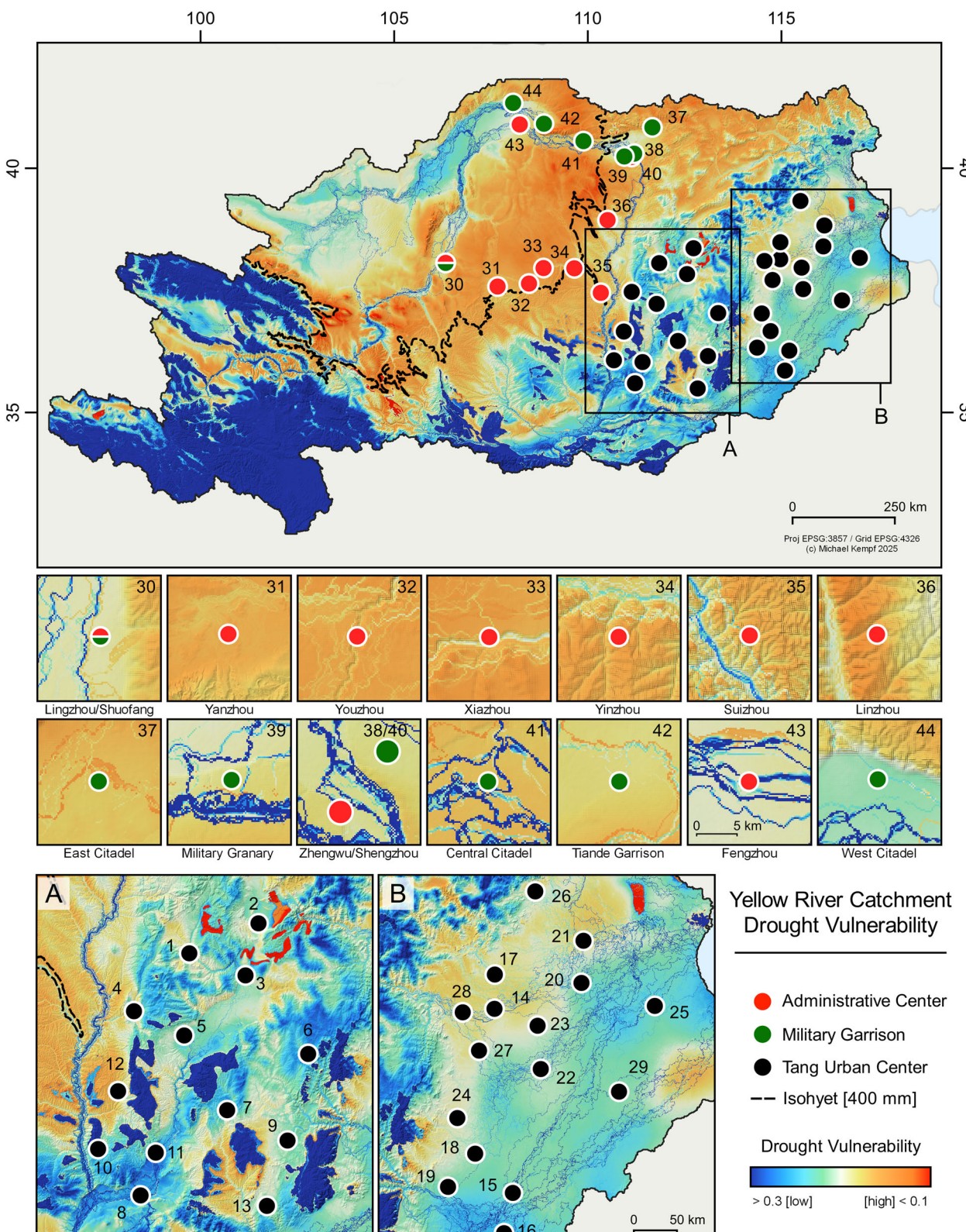

**Fig. 2 | Multiannual Yellow River hydroclimatic vulnerability model.** High values (blue) indicate low composite drought vulnerability; low values (red) indicate increased composite drought vulnerability (hydrological and agricultural drought). The drought vulnerability model is a composite of flow accumulation and the multiannual cumulative SPEI from 1980 to 2018 CE[91] (see supplementary Table S1 for site location and names and Figs. S4–S6 for data density plots). Administrative centers are given in red, military sites in green and Tang urban centers in black. **A, B** Urban center focus plots.

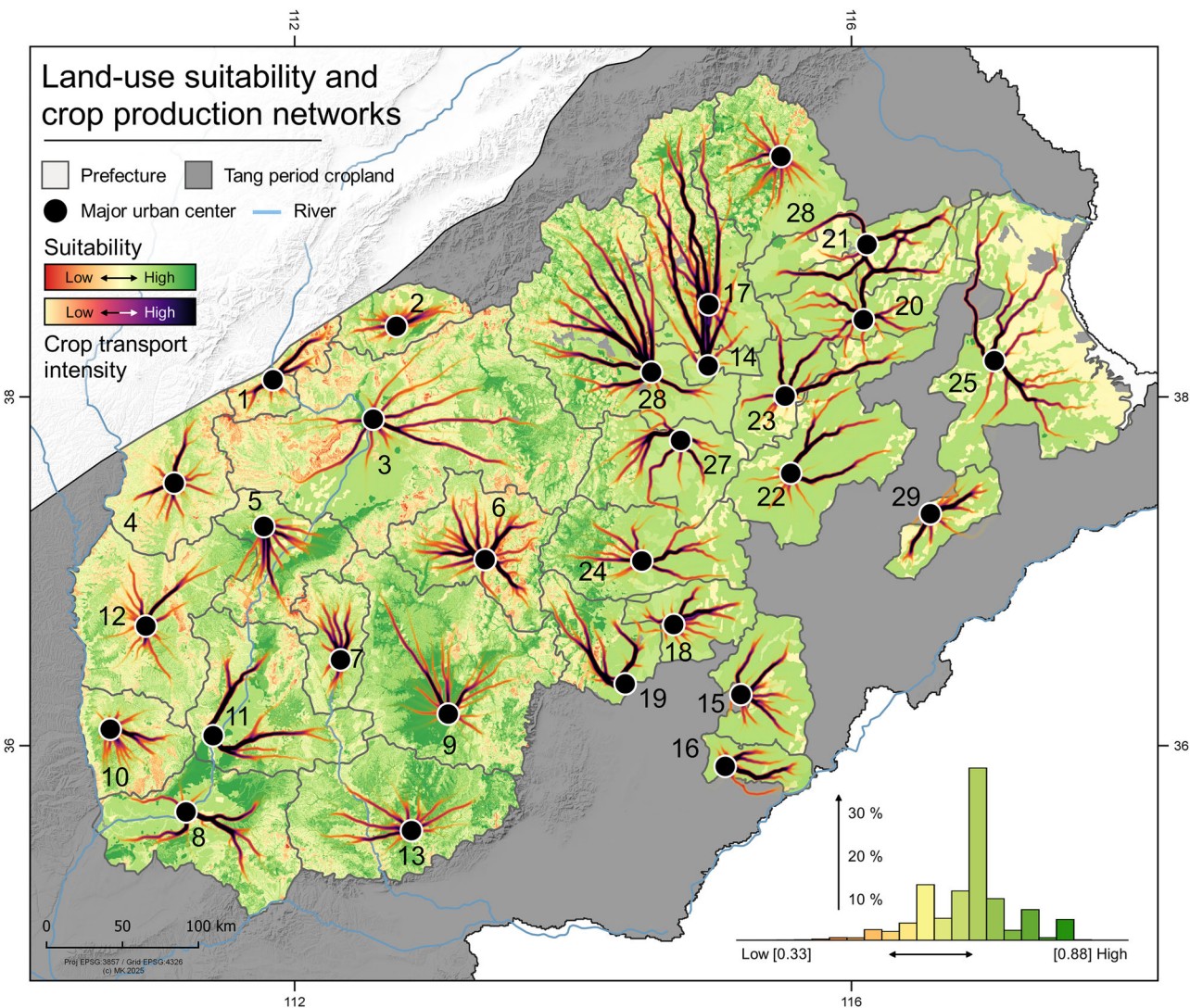

**Fig. 3 | Model-based simulation of land-use suitability during the late Tang Dynasty.** Light green polygon refers to the Tang Dynasty crop production area[82]. Land-use potential are the accumulative scores of soil properties[98,99], topography and climate variables[91]. The intensity corridors show the highest probability of transportation pathways towards each prefecture center. The bar plot shows the suitability score distribution in percent. Agricultural potential is generally above mean value of 0.5 across the late Tang Dynasty productivity zone (normalized standard scores from 0 to 1; reference period of the climate variables is 1980–2018 CE. Tang towns and prefecture data acquired from CHGIS[104,105] (refer to Fig. 1 and Table S1 for site names; see Fig. S13 for data distribution in each prefecture and Fig. S14 for the entire catchment).

Particularly, soil properties play a crucial role in the spatial distribution of productivity scores. Cambisols are mostly abundant on terraces in the middle and lower parts of the Yellow River. The lower alluvial plain is characterized by fluvisols and luvisols towards the northern edges of the lower drainage basin and the northernmost part of the Tang croplands. Fluvisols and cambisols are mostly used for irrigated and unirrigated cultivation. Locally abundant, shallow and weakly developed regosols and eroded leptosols have lower land-use potential, mostly in combination with steeper slopes and higher elevation (Figs. S9–S12). Solonchaks with high soluble salt content are characterized as waste land and cannot be cultivated[57]. Compared to land-use suitability, the modeled prefecture-level agricultural crop production pathways were estimated. Within each prefecture, the most likely pathways for crop production and transportation to the urban centers were simulated depending on topographic, hydrological and climatic variables (Fig. 3). These modeled pathway simulations (or scenarios) are a representation of the topographic accessibility and land-use suitability and show the connection of the Tang rural hinterland to the urban supply hubs from which centralized and organized long-distance networks started. Which is a key factor, as the state-controlled deployment of troops along the border, Tang population growth[53,71], and the establishment of administrative centers and military garrisons during the late Tang Dynasty pushed the local economic exploitation up too far and rapidly overstrained the crop productivity. The population development in the northern frontier zone was connected to military commands. The West and East Citadels became the administrative centers of standing armies (Tiande Garrison and Zhenwu Garrison) with the East Citadel being crucial for the defense of military granaries and the transport system connecting the northern garrisons to the agricultural heartland of the Tang[31,32,72,73].

To offset high transportation costs, the YRL garrisons partially relied on military farms operated by soldiers and state farms managed by farming families[74]. Local production avoided transport costs but required stable labor. The local climatic conditions and short growing seasons during colder and drier periods[40] limited grain output for the garrisons. However, efforts to boost production through irrigation measures and state-controlled farms largely failed. By the mid-8th century CE, military farms supplied only 20% of provisions, with the majority of grain and fodder for the armies relying on state purchases and long-distance transport[31,32].

## Climate-controlled crop supply networks

Late Tang Dynasty northern and eastern China experienced a strong shift towards drier and cooler conditions after 700 CE and during the 9th until the early 10th centuries CE (Fig. 4A–C)[75–79], challenging crop production particularly in the northern YRL. However, general trends towards lower rainfall were disrupted by frequent flooding events in the first half of the 9th century CE and between 880 and 890 CE. Such flooding events ($n > 1500$) are recorded in documentary sources over the last millennia with a maximum frequency of up to three events per year during dry and cool conditions[6,66]. Above normal run-off periods (Fig. 4D, E) in combination with increased land-use[6], deforestation[80], and consecutive drought years, also accelerated soil erosion, turning arable land into less productive land[81]. In the middle basin of the Yellow River, population growth during the Tang Dynasty exceeded the previous population numbers[6,82], which resulted in a strong transformation of the landscape and extensive forest clearing.

Grasslands prevailed due to widespread horse, caprine, and cattle breeding[83], but socio-political challenges starting from the mid-8th century CE led to an abandonment of the pastures. Later reclamation of the land and agricultural exploitation fueled soil erosion and increased the river sediment load at the expense of climate sensitive grasslands[6,84]. Eroded soils in the YRL, therefore, caused poor harvests, particularly under drier and cooler conditions of the late Tang Dynasty. Harvest grades during the first half of the 9th century CE, as reported by Hao et al. (2020), ranged from very poor to a maximum of average with a sequence of particularly poor harvests between 800 and ~840 CE[85]. These reduced yields were most likely caused by the interplay of population development, the cultivation of wheat instead of drought-resistant millet, land-use intensification, and soil erosion, and were accelerated by climatic pressure.

Climatic pressure increased during the late 9th and particularly the early 10th centuries CE. A rather humid period with above-normal run-off during ~880 and 900 CE increased the risk of flooding events in the middle and lower reaches of the Yellow River. After 900 CE, a rapid and strong decline in run-off coincided with a drop in precipitation (Fig. 4B, C, E) and the reconstructed EASM strength[75]. A weaker EASM will transport less rain to the northern and middle Yellow River[86] and increase regional hydrological drought. The end of the Tang Dynasty is then marked by a sharp decline in precipitation and run-off. Decreased precipitation and moisture availability could have affected upstream local crop production and downstream centralized agriculture as crop production beyond the irrigated core was reduced, and rain-fed farming was no longer possible. Consequently, crop production declined shortly before and around 900 CE with a sequence of very poor harvest grades[85].

In the northern military frontier zone, local crop production was not sufficient to supply the population and the military troops. Large-scale transportation networks were established to maintain crop supply from the Tang heartland to the frontier zone (Fig. 5, upper panel). Production in the fertile and less drought-prone Tang prefectures in the eastern part of the catchment and the North China Plain, however, must enter the supply networks by a highly cost- and time-effective transportation system (Fig. 5, middle panel). The reconstructed prefecture-level production ran via high probability network corridors towards each central supply hub from which organized distribution through a set of environmentally adapted and optimized least cost networks occurred. Depending on the climatic conditions across the northern YRL and the upstream catchment, dry or humid periods impacted the overall networks through potential flooding events or during years with increased drought vulnerability. Semi-arid areas became less favorable to travel through due to decreased water availability and higher risk of mortality of horses, cattle, and people. During humid periods, flooding events affected the pathways, making river crossings and river-based transport difficult or impossible. Land-routes were rather the preferred option of transportation with fresh water being abundant during the long travels (Fig. 5, lower panel).

The late Tang Dynasty accumulative networks show the most likely transportation corridors during 108 years of climatic variability and surface response. Each node represents the most cost-effective connection between the late Tang urban centers in the crop production zone, the frontier administrative hubs and the military garrisons. Late Tang decentralization, exemplified by warlords controlling key administrative zones, however, likely fragmented supply networks and reduced central oversight, challenging the accuracy of the modeled routes presented here[53,71]. Pathway density was calculated using a Kernel Density Estimate (KDE) of the connective strings. Each network string, however, has been calculated individually using the climatic conditions present during the respective chronological bin derived from the run-off index. The resulting 53 environmental response bins indicate the approximate impact of increased drought conditions and flooding vulnerability in the middle and lower reaches of the YRL. Each network was individually weighted with the respective run-off coefficient of the record, assigning it a specific magnitude of vulnerability. Areas that are characterized by semi-arid or arid conditions will become drier during drought periods whereas flood-prone areas are more likely to be flooded during peaks in run-off. For each bin, the network looks different, avoiding extended semi-arid areas during droughts as well as flood zones during wet periods. The KDE emphasizes network continuity during periods of climatic variability as well as the adaptation of pathways, particularly in areas that are, for example, more vulnerable to small-scale changes in moisture budget (e.g., the Mu-Us desert). The MST (Minimum Spanning Tree) underlying each network is created from minimum local travel time between the nodes (e.g., the urban centers) and shows the maximum traffic between the closest production hubs (high spatial proximity) in the eastern part of the Tang cropland. From there, crops were transported via the central hubs IDs 6 (Liao Zhou) and 9 (Shangdang Jun) and particularly from Taiyuan (ID 3) to the west and then to the newly established late Tang administrative centers. The model's representation of the transport network, however, may be oversimplified given the complex military fragmentation of Late Tang regional warlords (*fanzhen*). For instance, administrative centers 15–16 were controlled by the Weibo warlord, 20–21 by the Lulong warlord, and 22–23 and 27 by the Chengde warlord. These three warlords consolidated military, administrative, and fiscal authority, operating with substantial autonomy from the central government during much of the late Tang period.

Due to the lower spatial proximity of the administrative centers in the west, higher KDEs are observed, which furthermore scatter substantially due to environmental constraints underlying the network analysis. Here, the marginal zones of the Mu-Us desert with high aridity play a decisive role in network adaptation during periods of climatic variability. Marginal zones of semi-arid grasslands can be used as transportation zones, however, consecutive drought years, for example during the first decade of the 9th century CE and particularly the early 10th century CE, would push the margins further south, impacting the water and fodder availability and consequently altering the pathways. High travel times (Fig. 5, middle panel) further increase the vulnerability of travelling through semi-arid areas (Fig. 5, lower panel). In Fig. 5 (lower panel), the vulnerability of travelling across the landscape is visualized as an interpolated raster based on points sampled from the simulated connections between the sites. At each point, vulnerability values reflect the combined influence of flood and drought conditions along the underlying route segments, and when interpolated these values spread outward and overlap, producing broad corridors where exposure is consistently high. Lower travel times, higher spatial proximity, abundance of freshwater and positive CWB decreases the drought vulnerability and the risk of livestock and human loss (e.g., between the urban centers). The network vulnerability, however, is calculated cumulatively and equally incorporates the risk of travelling through areas with high flooding potential. Particularly, the northern YRL shows high vulnerability values, which is due to both semi-arid conditions during droughts and high flood risk during above normal run-off periods. The supply chains from the administrative centers to the military garrisons are the most vulnerable, caused by location in environmentally sensitive areas, low spatial proximity, and high travel times between the nodes.

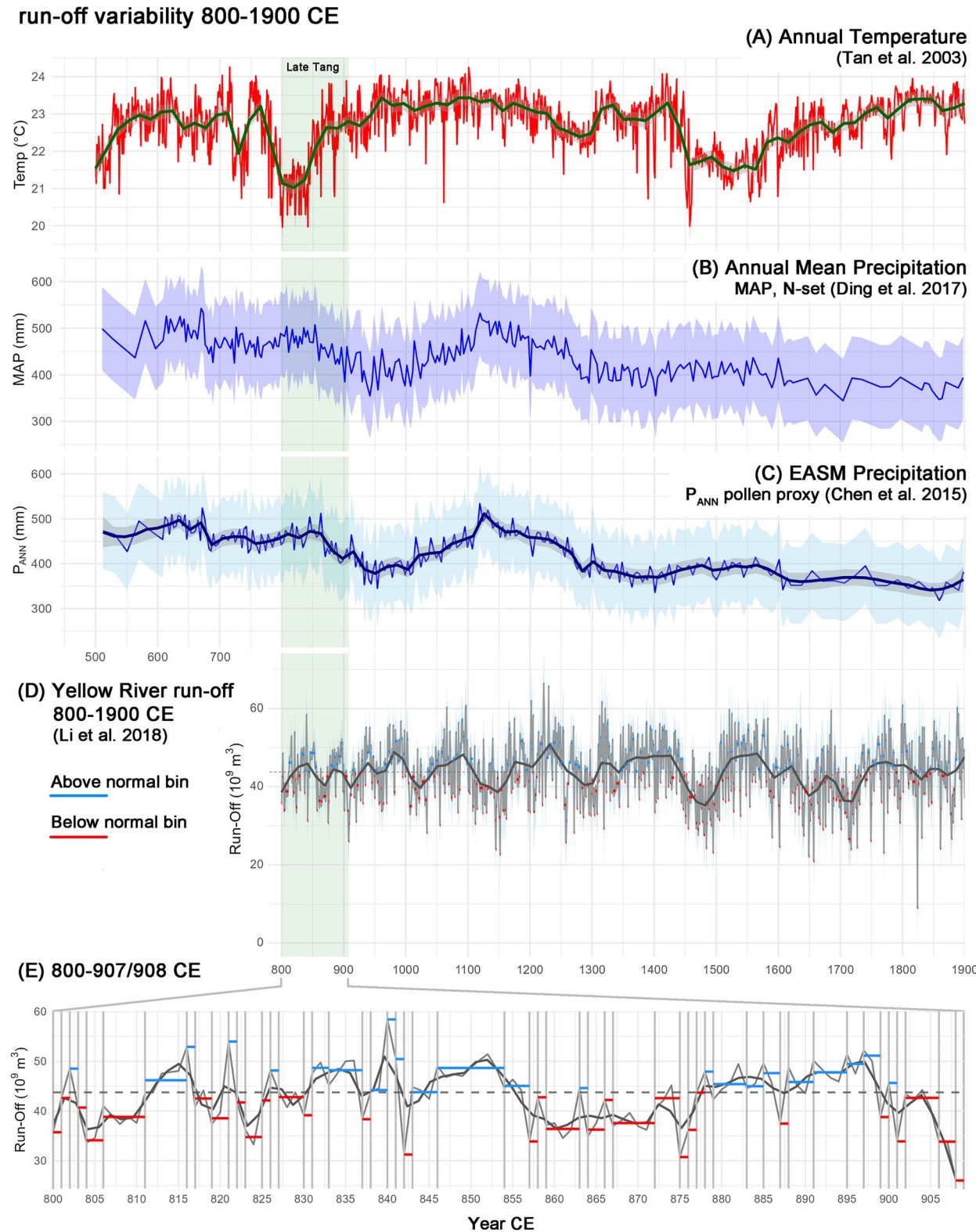

**Fig. 4 | Proxy-based reconstructions of climate and environmental changes in north-eastern China. A** Reconstructed annual warm season temperature at Beijing deriving from speleothem records[78]; **B** Annual mean precipitation and uncertainty in north-eastern China deriving from natural pollen data (MAP N-set[76,109]). **C** Pollen based reconstructed EASM precipitation and uncertainty (light blue shaded color area) with local regression (LOESS, grey shaded envelope) at Gonghai Lake (North China)[75]. **D** Reconstructed Yellow River run-off dynamics deriving from upstream tree ring records for the period 800–1900 CE[77]; **E** for the period 800–907/908 CE[77]. Using dynamic binning techniques based on data variance (see methods), 53 bins were determined between 800 and 907/908 CE that characterize above normal run-off (potential flooding) (blue) or below normal run-off (potential hydrological drought) (red) probability. Dashed line = general mean value; grey solid line = Locally Estimated Scatterplot Smoothing (LOESS, span= 0.07 (**D**, across the full record) and 0.08 (**E**, across the subset)).

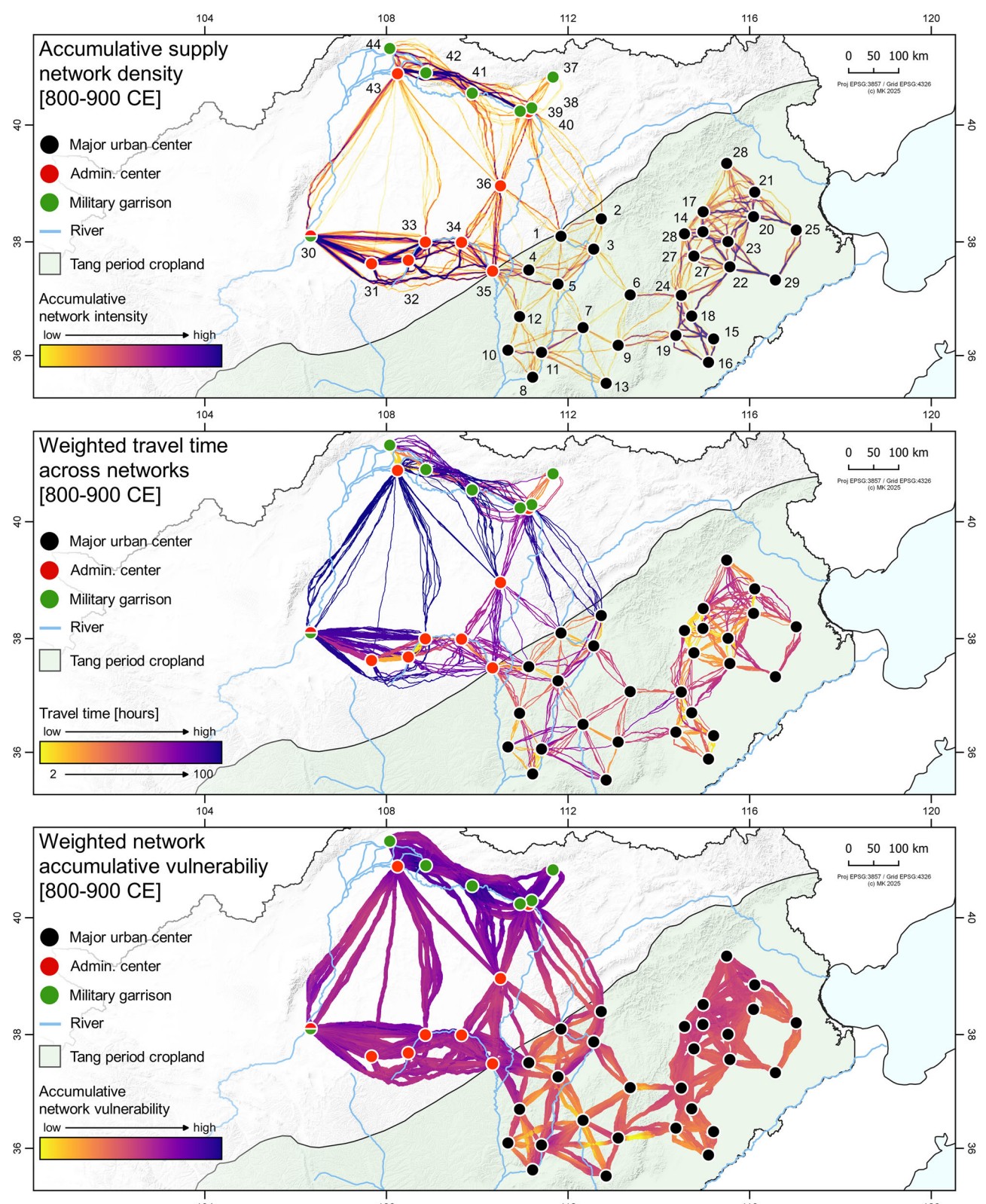

**Fig. 5 | Model-based simulations of Late Tang Dynasty supply network properties.** Upper panel: accumulative intensity for 53 climatically weighted network bins. Middle panel: travel time for each connection in the 53 bins in hours. Lower panel: IDW (Inverse Distance Weighting) interpolation of the accumulative network vulnerability to drought and flooding events across the bins.

## Conclusions and perspectives

This study provides a detailed, multi-faceted analysis of the factors that contributed to political instability during the final phase of the Tang Dynasty in China. It demonstrates how the interplay between hydrological droughts, flooding, and socio-political pressures contributed to and likely enhanced internal and external socio-political stress and instability that led to the collapse of the dynasty in the late 9th and early 10th centuries CE. To examine these dynamics, we developed a hydroclimatic landscape

vulnerability and network model centered at the northern YRL and the core agricultural regions of the late Tang state. This model simulates evolving food supply networks under non-stationary climatic conditions from 800 to 907/908 CE. The model shows that environmental stressors and logistical constraints shaped regional provisioning systems. Our findings point to three principal mechanisms of increased vulnerability:

(1) A transition in key crop production from drought-tolerant millet to more climate-sensitive wheat and rice increased the risk of harvest failure and famine.

(2) In response to pressure from invading groups originating from the Mongolian steppe, the Tang state established new military garrisons and administrative centers along its northern frontier. These sites faced chronic local food shortages and became dependent on increasingly fragile long-distance supply routes.

(3) Prolonged drought conditions in the northern YRL severely compromised the viability of these supply routes. Our simulations show that past land-based transportation systems consistently avoided the dry and fragile edges of the Mu Us Desert due to the environmental challenges these areas presented.

By modeling the vulnerability of historical food systems under climatic stress, this research highlights the critical role of environmental factors in the decline of state capacity and territorial cohesion during the final decades of the Tang Dynasty. This is the attempt to investigate, model and highlight the potential influence of environmental and climatic constraints on socio-political transitions and land-use strategies for this period in Chinese history. Overall, the study underscores the importance of integrating high-resolution climate data with spatial modeling techniques to better understand how hydroclimatic changes contributed to socio-political transformations in imperial China. The complex multicomponent model presented here is an innovative and likewise first attempt to integrate the various feedbacks and links of climate change, environmental response, and socio-political development. However, we clearly state that environmental challenges are contributing to social instability and that multifactor changes increase the vulnerability of a society to fail. We agree with previous research that pointed out the complex interplay of environmental shifts and Tang internal stressors[56]. Further applications are needed for other periods and areas to test the model assumptions, yet it is important to stress that this study investigates the late Tang Dynasty for itself and does not pretend that similar ecological settings would lead to similar effects on other socio-political systems. To the contrary, it is likely that socio-political issues intrinsic to the Tang Dynasty (e.g., choices related to not well adapted subsistence strategies) implied a lack of resilience given the environmental conditions during that period in this region. Moreover, the presented connectivity model can be complemented by different palaeoclimatic proxy data and other modeling techniques to actively integrate complex and fluctuating variables affecting landscape accessibility such as administrative capacity, state fiscal policy, military taxation, and the role of regional warlords in the political fragmentation.

## Material and methods
### Hydroclimatic drought vulnerability model
All analyzes were conducted using R software and open access data[87–89]. A hydroclimatic model for the northern YRL was created based on a digital elevation model (DEM, USGS: Global Multi-resolution Terrain Elevation Data 2010[90] (https://earthexplorer.usgs.gov/; last accessed 15th of January 2025) and climate variables[91] (CHELSA Version 2.1 (Climatologies at high resolution for the earth's land surface areas); https://chelsa-climate.org/downloads/, last accessed 15th of January 2025). The DEM was resampled to 250 m spatial resolution in EPSG:3857 using the *terra* package[92] in R software[87] and cropped to an operative bounding box (extent: 80°, 125°, 30°, 50°). Using the *whitebox* package[93,94], a streamflow network and the longest connection within the river network were calculated to estimate the extent of watershed of the upstream main stem (simulating the Yellow River)[51]. The DEM and all input variables were cropped to this window, which captures the catchment-inherent climatic conditions. The CHELSA monthly

resolved climate variables come in raster format ($n = 468$, ref. period 1980–2018) and were equally processed. Precipitation ($P$), average temperature ($T$), and PET as well as the climatic moisture index (CMI) were extracted for each month and cropped to the extent of the watershed of the Yellow River. From $P$ and PET, the CWB for each month was calculated as CWB[i] = $P$[i]-PET[i], scaled, and compared to the CHELSA CMI variable. The SPEI[95] index was processed, which is a multi-monthly pixel-wise estimate of drought conditions (*SPEI* package[96]). Particularly useful in semiarid areas or regions with a lot of water loss caused by evapotranspiration, the SPEI accounts not only for precipitation deficit but also for accelerated atmospheric evaporative loss (Fig. S9). Hydrologic modeling was carried out using SAGA GIS in the R environment via the package *Rsagacmd*[97]. We produced a run-off and flow accumulation grid from the successive flow routing algorithm with 10 iterations on the DEM (Fig. S10). The multi-annual SPEI composite was inverted and both the SPEI and the flow accumulation were normalized using a minmax function and summed up to the hydroclimatic vulnerability model (Fig. 2).

### Crop production probability model
We prepared a weighted Multi-Criteria Evaluation model for potential agriculture based on topography (elevation [m. asl], slope [°], $T$ [°C], CWB [kg/sqm/month] and soil units (250 m spatial resolution). All input variables were scored from 0 (low) to 1 (high) based on generalized affordances for agricultural purposes. We employed a soil classification workflow, which combines data from the Soil and Terrain database for China[98] with the internationally harmonized FAO soil taxonomy[99]. Climate variables (monthly T and CWB, 1980-2018) from CHELSA were subset to the growing period April–October and multiannual monthly mean values and growing season averages were calculated. Terrain analysis was performed on the DEM and the slope (in degrees) was calculated. The elevation and the slope were scored as shown in Table S2 with elevation up to 1000 m representing the most suitable range for crop production and higher values characterized by declining scores. Flat areas and gentle slopes were considered highest scoring whereas increasing slope gradients show increased risk of soil erosion and less manageable terrain. $T$ below 5 °C indicates non-suitability and the optimum $T$ corridor was set to between 11 and 23 °C. Higher values were considered less favorable due to increasing risk of crop failure and need for intensive irrigation, particularly in semi-arid or arid areas. Moisture availability was similarly handled with negative values being unsuitable for agriculture and the optimal corridor being towards the upper quartile of the mean. However, too much water availability was rendered useless due to increased waterlogging vulnerability.

For the soil scores, the FAO attributes were extracted and added based on their chemical and physical characteristics. For example, saline soils (*Solonchaks*) were interpreted as less suitable for crop production whereas alluvial soils (*Fluvisols*) or *Chernozems* were attributed to high fertility and agricultural potential. Using the soil "Qualifier" in combination with the soil unit, the scores were further differentiated, for example into a *Salic Fluvisol*. Finally, all variables were stacked and weighted using a linear combination as

$$\text{Suitability} = a * \text{Elevation} + b * \text{Slope} + c * \text{CWB} + d * \text{Temperature} + e * \text{Soil}$$

with $a = 0.15$, $b = 0.15$, $c = 0.4$, $d = 0.1$ and $e = 0.2$. The CWB was given a high weight compared to T and terrain characteristics to enhance the power of moisture availability, particularly in the semi-arid northern parts of the study area. The result is a continuous suitability index (0–1), where values close to 1 indicate high agricultural potential under climatic, topographic, and soil characteristics (Figs. 3, and S1–S6).

### Yellow River run-off coefficient
Using the Yellow River streamflow reconstruction (800–2000 CE[77]) based on the tree ring data, a catchment-wide drought and flooding vulnerability model has been worked out. The reconstructions highlight flooding events

as well as flow reductions linked to prolonged droughts, particularly in the upstream area of the YRL. To focus on pre-modern natural flow patterns, the annually resolved dataset is first subset to the period between 800 CE and 1900 CE, excluding the impacts of recent human-induced environmental impacts such as dam constructions and water management policies that began to dominate after around 1960[77]. We calculated a restrictive variability threshold based on statistical properties of the year-to-year streamflow differences to reclassify into periods of consistent hydrological characteristics. The combination of the mean and standard deviation (mean-SD/2) of these differences serve as boundaries for dynamic bins, each representing a distinct period of relative uniformity in streamflow. Because a rather restrictive threshold was defined, the binning results in a plethora of bins rather than merging periods with significant variability into chronologies of stable conditions. Eventually, the data was subset to the period of interest from 800 to 907/908 CE (Fig. 4).

## Site connectivity models

For the least cost path (LCP) analyzes between administrative sites and military fortifications, three different connectivity models were generated: All single administrative sites were used to calculate individual connectivity patterns, followed by the military sites and a general landscape permeability model based on catchment-wide connectivity (FETE approach, from everywhere to everywhere)[100,101] (Figs. S7 and S8). The first step of separating military from administrative sites allows for considering potential effects of the diverse logistics and socio-political motivations or constraints related to movements in each specific context, which are otherwise variables that are difficult to quantify. The DEM was used as an input variable to estimate slope and terrain permeability within the window of operation (the catchment). Second, a custom friction model was created based on the DEM, the flooding vulnerability model (flow accumulation), and a regional coefficient that characterizes all areas with significant aridity based on the aggregated multiannual CWB and a threshold of mean+SD/2 across the catchment. In the model, high slope gradients, high flooding vulnerability/high probability of branching rivers, and high aridity were estimated as less likely for movement, hence a low probability of accessibility was given. Using the *r.walk* function in the GRASS GIS interface in R via the package *rgrass*[102], LCPs were generated between a set of regularly sampled points ($n = 30,000$) across the catchment and the target points of military or administrative centers[100,101]. From each generated LCP, regular points were sampled at an interval of 1000 m. Using operational chunks due to memory efficiency, KDE were carried out (*spatstat* package[103]) on the points with a Gaussian Kernel, a sigma of 1000 m and a resolution of 250 m. Finally, all KDEs were summed up to a high probability movement corridor model for military and administrative centers. In a last step, a general connectivity model (FETE) has been worked out, in which target and starting points were taken from the same regularly sampled point pattern with a sample size of $n = 1289$, resulting in over 1.6 million LCPs across the catchment area. The model visualizes the distribution of potential high movement probability based on terrain, simulated riverbeds, and high aridity and serves as a comparison dataset to test the connectivity patterns of the observed Tang Dynasty sites (Figs. 5, and S7 and S8).

## Crop production and supply chain network model

Using similar LCP techniques as previously described, a large-scale supply network between major Tang Dynasty towns and the newly established administrative hubs and military garrisons in the north has been worked out. The additional Tang towns were extracted from CHGIS[104,105] (last accessed 05th of April 2025). Both datasets were filtered for chronological start and end year to select only occurrences that already existed in the 9th century CE and until the end of the Tang Dynasty in 907 CE. Point data and polygons were cropped to the watershed of the Yellow River and the Tang Dynasty crop production area reported from[82]. Using a spatial join, we further selected only those prefectures that had an existing town record and only those towns that were located within a prefecture. A total of 31 towns represents the Tang Dynasty sample. Using a loop in R, the *rgrass* package

and the method described above, we sampled $n = 5000$ points within each prefecture polygon and created an LCP collection for each point in $n$ targeted at the central town based on the friction raster described above. We equally produced the LCP density with sigma=1000 and sampled points each 1000 m from each linestring (Fig. 3).

**Least cost network estimates.** We produced a set of weighted FETE LCP collections from the merged administrative sites in the north and the Tang cities. For this, we prepared 53 different input friction surfaces, which were created as described above, however, using the dynamically binned run-off values as coefficients ($n = 53$). The friction is the combination of normalized topography (DEM), moisture availability (CWB, arid areas), and flooding potential. We normalized the binned mean run-off values for each dry or humid bin and rescaled them to between –0.5 and +0.5 for dry and humid, respectively. Finally, the coefficient was applied to the climatic input variables of the friction as weight. For dry bins, the CWB will be reduced by the factor*1/20 to simulate increased drought conditions. Humid bins enhance the CWB by the same ratio. Finally, the arid areas below the threshold mean+SD/2 of the CWB were calculated for each weighted bin. The flow accumulation was equally weighted to enhance flooding potential during humid bins and reduced flooding during dry bins. This results in 53 different FETE LCP collections of crop supply centers and northern border administrative centers depending on environmental variability during the 9th and early 10th century CE Tang Dynasty. Equally, we computed the weighted LCPs between the administrative centers and the military garrisons.

**Network travel time estimates.** From these LCP collections, we created 53 networks for crop production centers targeted at the administrative centers and then from there to the military sites via the *igraph* package in R[106]. We did not estimate Euclidean distance or the length of the calculated LCPs but estimated the energy expenditure of travelling along the LCPs represented in travelling time. We used the individual accumulative friction raster for each LCP subset, created the slope in radians and applied Tobler's hiking function[107] as

$$Speed = 6 \cdot e - 3.5 \cdot |slope + 0.05|$$

Each LCP linestring was sampled at regular 250-m intervals. At every sampled point, we extracted the slope from the raster to determine local walking speed. Using these speed values, we calculated segment-level travel times and summed them up to get the total estimated time (in hours) for each linestring in each of the 53 LCPs for both collections. An undirected network graph was created where edges were weighted by travel time. To identify the most efficient structure that connects all sites with the least total effort, we computed a MST that connects all nodes without cycles, minimizing the sum of edge weights (travel time). The resulting networks represent those connections that would have offered the most efficient connectivity under the assumption of climatic, topographic, and hydrologic constraints for each binned chronological subset. The most cost-effective network corridors were estimated using a KDE (sigma=1000 m, eps=250 m, gaussian kernel) on the entire collection of network linestrings from the crop production centers to the administrative hubs to the military outposts. We sampled each linestring at 250 m, split the networks into segments, calculated the segment midpoints and applied the *density.ppp* function from the *spatstat* package as described above (Fig. 5, upper panel). The total travel times for each connecting string in the networks were added in Fig. 5, middle panel.

**Network vulnerability estimates.** We further estimated the most-vulnerable/least-sustainable network corridors across the period 800–907/908 CE. Using the sampled midpoints at 250 m distance, we extracted flood risk (successive flow accumulation grid) and drought vulnerability (accumulative CWB grid) raster values. To reduce noise and simulate

edge effects of climate influence on larger segments of the networks, the values were smoothed along the linestrings using a rolling mean window with $r = 100$ cells on each side of the midpoint. Flood and drought smoothed values were normalized, a "flood vs drought bias" metric was computed as norm(flood[i]) - norm(drought[i]). The difference was centered and rescaled to the interval $[-1,1]$ with $-1$ indicating strong drought dominance and 1 indicating strong flood dominance. A total vulnerability metric was computed by summing normalized flood and drought vulnerability scores. To visualize the result (Fig. 5 lower panel), the total cumulative vulnerability of all 53 weighted networks was computed as an Inverse Distance Weighting interpolation (IDW) via the *gstat* package[108]. The resulting cumulative vulnerability map represents long-term transport network resilience and exposure under dynamic hydrological stress across the late Tang dynasty landscape.

## Data availability
All data underlying the analyses are freely available from the internet and the sources are cited in the text. Auxiliary data and related files are available from this repository: https://doi.org/10.5281/zenodo.15688117[88].

## Code availability
Code is available from this repository: https://doi.org/10.5281/zenodo.17151487[89]. File *Code_hydroclimate.R* produces the hydroclimate vulnerability model, the SPEI index and the density plots of the supplementary material. File *Climate_panel_figure.R* produces the panel figure. File *Code_land_use_probability.R* produces the land-use probability map. The networks and least cost densities are split into several steps: *LCP_code_F-ETE.R*; *LCP_code_ADMIN.R*; *LCP_code_MILITARY.R* form the supplement least cost figures. The code *LCP_code_TANG_within_prefectures.R* produces the LCPs within the prefecture and the network among the sites. *LCP_code_ALL_sites_and_networks_climatically_weighted.R* is the code that provides an example for a climatically weighted LCP and MST using the first of 53 climate bins.

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

## Acknowledgements

M.K.'s research at Basel and Cambridge is funded by the Swiss National Science Foundation (SNSF/SNF): Project EXOCHAINS - Exploring Holocene Climate Change and Human Innovations across Eurasia (SNSF grant number: TMPFP2_217358). Open access publication is funded by the Swiss National Science Foundation (SNSF/SNF) and M.K.'s project EXOCHAINS. E.X. acknowledges support from the ERC Synergy project EUROpest under Grant Agreement number 101166700. F.C., E.X., J.L., M.F., acknowledge the Association for Trans-Eurasia Exchange and Silk-Road Civilization Development (ATES). We thank Zhuangzhuang Bai for comments.

## Author contributions

M.K. designed the study, generated the code, performed the analyses, created the figures, and wrote the paper with input from M.D., R.S., M.F., F.C., J.L., E.X., and U.B. All authors consent to the final version of the paper.

## Competing interests

The authors declare no competing interests.
