## [Transparent Peer Review file · Communications Earth & Environment]

Hydroclimatic instability accelerated the socio-political decline of the Tang Dynasty in northern China

Corresponding Author: Dr Michael Kempf

Version 0:

Decision Letter:

Dear Dr Kempf,

Your manuscript titled "Hydroclimatic extremes in northern China contributed to the decline of the Tang Dynasty" has now been seen by 3 reviewers, and we include their comments at the end of this message. They find your work of interest, but some important points are raised. We are interested in the possibility of publishing your study in Communications Earth & Environment, but would like to consider your responses to these concerns and assess a revised manuscript before we make a final decision on publication.

We therefore invite you to revise and resubmit your manuscript, along with a point-by-point response that takes into account the points raised.

In particular, for publication in Communications Earth & Environment we request that you provide (1) a comprehensive overview of related literature, (2) criteria for extreme events, and the comparison of models across different historical periods, and (3) discuss the impact of social and political changes and possible limitations due to excluding these factors from the model.

Please highlight all changes in the manuscript text file.

Please submit your point-by-point responses as a separate file, distinct from your cover letter where you can add responses to the Editors' comments that you do not want to be made available to the reviewers. Word files are preferred. We recommend that any figures, tables or graphs that are included in the response to reviewers are also included in the main article or Supplementary Information.

Please use the following link to submit your revised manuscript, point-by-point response to the referees' comments (which should be in a separate document to any cover letter), a tracked-changes version of the manuscript (as a PDF file) and the completed checklist:

Link Redacted

We hope to receive your revised paper within six weeks; please let us know if you aren't able to submit it within this time so that we can discuss how best to proceed. If we don't hear from you, and the revision process takes significantly longer, we may close your file. In this event, we will still be happy to reconsider your paper at a later date, as long as nothing similar has been accepted for publication at Communications Earth & Environment or published elsewhere in the meantime.

Please do not hesitate to contact us if you have any questions or would like to discuss these revisions further. We look forward to seeing the revised manuscript and thank you for the opportunity to review your work.

Best regards,

Ola Kwiecien, PhD
Editorial Board Member
Communications Earth & Environment
orcid.org/0000-0001-6018-9181

Martina Grecequet, PhD
Senior Editor
Communications Earth & Environment

EDITORIAL POLICIES AND FORMATTING

- Behavioural and social science
- Ecological, evolutionary & environmental sciences
- Life sciences

Furthermore, please align your manuscript with our format requirements, which are summarized on the following checklist: <https://www.nature.com/documents/commsj-phys-style-formatting-checklist-article.pdf> Communications Earth & Environment formatting checklist

and also in our style and formatting guide <https://www.nature.com/documents/commsj-phys-style-formatting-guide-accept.pdf> Communications Earth & Environment formatting guide .

*** DATA: Communications Earth & Environment endorses the principles of the Enabling FAIR data project (<http://www.copdess.org/enabling-fair-data-project/>). We ask authors to make the data that support their conclusions available in permanent, publically accessible data repositories. (Please contact the editor if you are unable to make your data available).

All Communications Earth & Environment manuscripts must include a section titled "Data Availability" at the end of the Methods section or main text (if no Methods). More information on this policy, is available at <http://www.nature.com/authors/policies/data/data-availability-statements-data-citations.pdf>.

If a community resource is unavailable, data can be submitted to generalist repositories such as <https://figshare.com/> or <http://datadryad.org/> Dryad Digital Repository. Please provide a unique identifier for the data (for example a DOI or a permanent URL) in the data availability statement, if possible. If the repository does not provide identifiers, we encourage authors to supply the search terms that will return the data. For data that have been obtained from publically available sources, please provide a URL and the specific data product name in the data availability statement. Data with a DOI should be further cited in the methods reference section.

REVIEWER COMMENTS:

Reviewer #1 (Remarks to the Author):

Crop circulation and food security remain fundamental challenges intimately tied to human survival and require serious global attention. Historical strategies for crop procurement, storage, and transportation, as well as climate–society interactions during the Tang Dynasty, can offer valuable insights for building more resilient food systems today. This study introduces a multicomponent hydroclimatic vulnerability model for crop supply networks in northern China, offering new insights into how climate variability may have contributed to the decline of the Tang dynasty. While the study offers valuable insights at the regional level, it falls short in addressing the broader context of food supply across the Tang dynasty's full territorial span. Additionally, the simplified modeling approach overlooks important socio-political dimensions that likely played a crucial role in shaping the empire's food security and adaptive capacity. Substantial revision would be necessary before the manuscript could be considered for publication. I strongly recommend the authors pay more attention to the following critical concerns from historical scholarship and argumentative logic to help strengthen the paper.

1. The central argument of the paper appears to be that environmental changes in northern China significantly contributed to the decline and eventual fall of the Tang dynasty. While the authors provide substantial environmental data to support this claim, the connection to the broader historical processes remains underdeveloped. The least-cost path analysis employed to reconstruct transportation networks is technically impressive; however, it potentially underrepresents the complexity of actual historical logistics during the late Tang period. As the article states that the model focuses primarily on environmental variables such as climate, topography, and soil, while excluding critical socio-political factors like population distribution, administrative capacity, state fiscal policy, military taxation, and the degree of political fragmentation (e.g., the role of regional warlords). These elements would have significantly influenced both local agricultural productivity and the viability of long-distance food transport during the late Tang period. In particular, the manuscript would benefit from a more explicit discussion linking the environmental shifts with contemporaneous political instability, local uprisings, and demographic changes during the late Tang period. For example, basic population data—an essential indicator of socio-political change—is entirely absent. Moreover, it is worth discussing whether the supply path established by the model actually exists, or to what extent it represents the "potential path" rather than the "actual path" at that time.
2. The manuscript does not sufficiently engage with the dominant scholarly narrative about the fall of the Tang dynasty. Most historians emphasize the financial and political reliance of the Tang government on the southeastern provinces in its last years. The disruption of communication and control between the central government and southeastern provinces during large-scale rebellions is widely regarded as a key factor in the dynasty's collapse. If the paper seeks to argue for the critical importance of northern environmental changes, it should at least acknowledge and address these prevailing interpretations to clarify its contribution to the existing literature. Besides, while the focus on the Yellow River Loop region is clearly justified in terms of its strategic and military relevance, the analysis overlooks major grain-producing regions—in particular, the southeastern region emerged as a crucial agricultural hinterland during the late Tang period, characterized by a rice-based agrarian system with markedly different climatic responses and seasonal sensitivities compared to the millet–wheat regime of northern China. Notable agricultural development in southern China has been recognized in the Late Tang dynasty—partly due to the introduction of wheat enabled the cultivation of water-scarce highlands and hills in the southern areas, along with the adoption of rice-wheat crop rotation. Northern single-crop farming systems limited both land-use efficiency and overall grain output. These structural differences in agricultural practices should be considered when assessing regional vulnerability to environmental stress. This study, however, does not sufficiently explain why this important southern region is omitted from the modeling framework, nor does it assess how its inclusion might alter the results.
3. The authors argue that the shift from drought-resilient millet to more climate-sensitive wheat during the late Tang increased the regional vulnerability to hydroclimatic stress. While this interpretation is broadly supported by some archaeobotanical and isotopic studies, the spatial and temporal uniformity of this crop transition remains uncertain. In particular, the claim that wheat had largely supplanted millet across the northern and southern part of the Yellow River Loop during the late Tang period requires further substantiation. The growing season of different crops should also be taken into account. Wheat is predominantly a winter crop in northern China, while the climate proxies used in this study focus on summer/annual precipitation and summer temperature, which may not adequately reflect the hydrothermal conditions that determine winter wheat productivity. Relying on summer-biased climate reconstructions to infer wheat yield sensitivity may lead to misinterpretations.
4. The manuscript emphasizes the role of hydroclimatic extremes (floods and droughts) in shaping the decline of the Tang Dynasty. However, the criteria used to define "extreme events" remain unclear. In several sections, events are labeled as "extreme" simply because they deviate from the mean, which does not necessarily meet a climatological or statistical threshold for extremity. While the reconstructed precipitation records indicate a general drying trend during the late Tang period, they do not clearly show the occurrence of exceptionally severe droughts. Similarly, although runoff records suggest a marked decline between 900–907 CE, the classification of this period as an "extreme drought" lacks a defined baseline.
5. The paper would be significantly strengthened by incorporating relevant Chinese historical sources. Given that the topic deals with historical periods for which abundant textual literature exist, it is problematic that the manuscript does not cite or discuss any Chinese textual records. This omission raises concerns about the historical grounding of the paper's argument.
6. I'm wondering whether the vulnerability model and the production probability model vary across different historical periods? It would be valuable if the manuscript includes any comparative analysis of how these models differ under varying climatic contexts. Such a comparison could strengthen the argument by showing how Tang societies adapted—or failed to adapt—to environmental stress over time. Otherwise, this type of reasoning can be problematic, as similar environmental patterns could potentially be found in other periods (e.g., the early Tang), without leading to the same historical outcomes. A more nuanced and critically reflective approach would improve the analytical rigor of the paper.

Overall, the manuscript demonstrates valuable data synthesis and cross-disciplinary potential, but it needs a more robust integration of historical analysis and engagement with established historiography to make a convincing case.

There are some other suggestions:

1. The paleoclimate reconstruction sites used in the study should be clearly marked on the corresponding figures, like Fig 1.

2. In Figure 1, it would be helpful to include additional historical context by labeling the Tang dynasty capital, major administrative centers, key granaries (grain distribution hubs), and other relevant background information. These additions would enhance the reader's understanding of the geographic and political landscape referenced in the manuscript.

3. Fig. 1: "annual average temperature (C) and precipitation (D) (Karger et al 2017)", the meaning of the "(>+15) and (<-15)" shown in panel C is unclear, the authors should explicitly explain what this threshold represents.

4. Fig. 2: the meaning of each panel should be clearly indicated in the figure caption. Additionally, the symbols or data points shown in the figure should be explained in the caption.

Reviewer #2 (Remarks to the Author):

The authors study the interplay between environmental and hydroclimate conditions, cultural and societal stress, food production and transport, and infrastructure vulnerability. The study is very interesting and a very good example of interdisciplinary research. It helps to understand the growth and limitations of urban centers during the Tang dynasty in China in the light of climate variability, which is novel and highly original research. It will also be of high interest to a broader community and will help to further establish quantitative methods in this field. I strongly suggest publication of this study.

I have only a few comments which will help to further improve the paper (and its understanding). Most of them are just minor.

In Fig. 1: I suggest to flip the elevation colour bar (low at the bottom).

Fig. 2: It is unclear why the drought vulnerability index has small values for high vulnerability and high values for low vulnerability. From a logical point of view, it would make more sense to have large values for high vulnerability.

Fig. 3: The colours are not easy to see. In particular, the green shaded area of the "Tang period cropland" is not visible. The colours of "Suitability" and "Crop transport intensity" are not easy to assign in the map as the colours are the same.

Fig. 4: In panel (C), there are two shaded areas. Meaning unclear. In the caption: "... characterize below normal run-off (potential flooding) (blue) or above normal run-off ..." – below normal and above normal seems to be swapped.

Line 335pp: add references.

Fig. 5 (and related results discussed in the text): It is unclear what the different line widths of the links mean in the lower panel means. Perhaps there is no meaning? But then I wonder why the links are thicker. I also suggest to increase the line width of the links in the other two panels in general, as they are quite thin.

Is there any evidence from local sites for these road networks? I suppose that it might be possible to find some artifacts or other remnants along the ancient roads?

Abbreviation IDW should be explained (is only explained at the end of the manuscript in the methods section).

Lines 475pp: Is the suitability equation some standard approach? I wonder why it is a simple linear combination. Any evidence for this? And how to select the coefficients (aside of coefficient c). And how was it ensured to have a value range between 0 and 1? From the equation it is not clear (and could have larger values, I guess).

Line 562: I suggest to use italic font only for the variables. The italic style for the sign of absolute values looks more like a sign for division. At first I was confused.

Line 902: I suppose the figure references are not correct and should be Figs. S15 and S16, where they are showing not administrative and military sites in the different figures but the connectivity (S15) and permeability (S16).

Fig. S16: Should there be differences in the lines shown between the upper and the lower panel? I cannot see any difference (only admin and military sites are different).

Reviewer #3 (Remarks to the Author):

Kempf et al. set out to better understand and explain the role of climatic variation on the Tang Dynasty's 9th and 10th century CE political fortunes by focusing on the intersection of hydrological variance, overall moisture availability, and land use in the Yellow River Loop region of northern China. In order to achieve this, the authors developed a 'multicomponent hydroclimatic vulnerability' model that draws on environmental variables, land use patterns, and historical processes such as network adaptation.

Overall, the paper manages to tie multiple threads of evidence together, neatly explaining how political and land-use decisions can interact with both naturally and anthropogenically variable climate cycles to engender famine and political instability (and/or sustainability) in premodern societies.

The metamodel that the authors developed and used to synthesize the various factors assessed in the study is novel in both its specifics and its general purpose, as is the interpretation that they draw from the model's output. Indeed, while multicomponent models are now becoming key means for untangling and interpreting the complexities involved in

premodern sociopolitical change, this particular approach is effective in addressing both the essential agency and passive impacts in human responses to environmental factors.

Untangling, presenting, and then recombining elements is no simple feat, but developing and deploying such an effective model produces a convincing argument that should push forward the field of environmental archaeology as well as how archaeologists, anthropologists, and social historians approach past social change. Given the clearly defined datasets, clear methodology, and code availability, sufficiently skilled researchers should be able to reproduce this work.

** Visit Nature Portfolio's author and referees' website at www.nature.com/authors for information about policies, services and author benefits**

Communications Earth & Environment is committed to improving transparency in authorship. As part of our efforts in this direction, we are now requesting that all authors identified as 'corresponding author' create and link their Open Researcher and Contributor Identifier (ORCID) with their account on the Manuscript Tracking System prior to acceptance. ORCID helps the scientific community achieve unambiguous attribution of all scholarly contributions. You can create and link your ORCID from the home page of the Manuscript Tracking System by clicking on 'Modify my Springer Nature account' and following the instructions in the link below. Please also inform all co-authors that they can add their ORCIDs to their accounts and that they must do so prior to acceptance.

Version 1:

Decision Letter:

Dear Dr Kempf,

Your manuscript titled "Hydroclimatic instability accelerated the socio-political decline of the Tang Dynasty in northern China" has now been seen by our reviewers, whose comments appear below. In light of their advice we are delighted to say that we are happy, in principle, to publish a suitably revised version in Communications Earth & Environment.

We therefore invite you to edit your manuscript to comply with our format requirements and to maximise the accessibility and therefore the impact of your work.

EDITORIAL REQUESTS:

*****Please take care to match our formatting and policy requirements. We will check revised manuscript and return manuscripts that do not comply. Such requests will lead to delays. *****

SUBMISSION INFORMATION:

OPEN ACCESS:

Communications Earth & Environment is a fully open access journal. Articles are made freely accessible on publication. For further information about article processing charges, open access funding, and advice and support from Nature Portfolio,

please visit <https://www.nature.com/commsenv/open-access>

Link Redacted

Best regards,

Ola Kwiecien, PhD
Editorial Board Member
Communications Earth & Environment

Martina Grecequet, PhD
Senior Editor,
Communications Earth & Environment
Consulting Editor,
Communications Sustainability

REVIEWERS' COMMENTS:

Reviewer #1 (Remarks to the Author):

The authors have answered all my questions in the revised version. I think it could be accept now.

Reviewer #2 (Remarks to the Author):

The authors addressed all my comments professionally and satisfactorily. I have no additional feedback and recommend accepting this significant study.

Reviewer #3 (Remarks to the Author):

The authors have sufficiently addressed all of the reviewers' comments and suggestions and this article appears to be ready for publication.

** Visit Nature Portfolio's author and referees' website at <http://www.nature.com/authors> for information about policies, services and author benefits**

Response to reviewers

Dear Reviewer,

We would like to thank you for your constructive comments and suggestions, which significantly improved the quality of the revised version R1.

In the following, please find a point-by-point response for all issues raised during the review process.

We thank you again for taking your time to review the manuscript and for your careful considerations that helped us to improve the paper.

With best regards,

Michael Kempf, on behalf of all authors

Cambridge, the 19th of September 2025

—

Reviewer 1

Dear Reviewer #1,

Thank you for your critical evaluation of our manuscript. We received three reviews and all of them identified disparate points, valuing specific parts of the model we developed as an exploratory tool to add potential drivers of socio-cultural changes connected to environmental factors. We are aware of the fact that working highly interdisciplinarity is not always an easy task and that there are many small factors that one can take into account when creating a model, however, a model, in its very sense, is a set of characteristics that describe a certain phenomenon or another characteristic. In this case, we deploy multicomponent environmental models as emulators to describe, not explain, land-use change and agricultural strategies of a historical society. Given that we cannot phenomenologically examine the Tang period itself, we have to use proxies for this. These can be written sources, as you point out, or simulations of climate feedback, both are ultimately indirect descriptions of potential realities that we cannot access directly. Hence, our data, historical as well as environmental, are indicators rather than realities. For example,

population data can never be accurate and is an assumption and a representation of something different. It is, *sensu stricto*, a model of the actual population. We are very aware of these facts, however, we tried to suggest a model that can contribute to future critical discussion and opens-up a debate that the scientific community can then evaluate.

In the following we discuss the individual points you raised and how we solved the crucial debate on socio-cultural factors in evaluating Tang period land-use and network models.

POINT 1

To address point 1 raised by the first reviewer, we added references to demographic data in the paragraph in which population distribution is actually already discussed. We also added an explanation in the method section on the role of socio-political constraints or motivations on these LCP as being part of the division between military and administrative centers - as these are otherwise difficult to quantify. Yet we acknowledged in the conclusion that several aspects raised by the reviewer would impact the model and that an agent-based model might allow for integrating these complexities in a future research step. As for the requirement to clarify the question of “potential” versus “actual path”, we would like to stress that this information is already present in the text (e.g. “the most likely pathways for crop production and transportation to the urban centers were simulated depending on topographic, hydrological and climatic variables (Fig. 3). These pathways are a representation of the topographic accessibility.

POINT 2

Thank you for this point. We have included a more thorough review of the Tang internal political developments. We are sure that these were the key factors, as you stated. Our model is solely complementary, meaning that we point out multiple stressors that could have contributed simultaneously to the decline of a political system. The data we used, which is Open Access data from Harvard to facilitate reproducibility, shows the agricultural hinterland as displayed in our figures on land-use suitability. We fully agree that wheat cultivation is the crucial factor here, as you said.

POINT 3

Thank you for this point. Tree ring data is seasonally informing climate conditions during the growing season. Winter wheat also has a high water demand during summer and particularly early summer, otherwise yields are critically at risk and crop failure can be the result of drought episodes during JJA. We selected annually resolved climate proxies to feed into the model, referring also to the availability and regional heterogeneity of the proxy data (see point 4). However, we acknowledge your critical reflections on seasonality and referred to that in the limitations of the study. As we note, it is likely that crop rotation cycles were implemented across much of the Tang Empire, with millet and wheat in the north and rice

and wheat in the south. Given that winter or monsoonal precipitation would have been high, the limiting factor would have been summer cropping, notably of millets, legumes, and vegetables. Across much of the northern empire, this would have mandated irrigation.

POINT 4

We thank the reviewer for this crucial point. We acknowledge that the criteria for defining "extreme" events, particularly for periods like the Tang Dynasty where direct statistical baselines are unavailable, should be clarified. We agree that labeling events as extreme based solely on a deviation from the mean is insufficient. Our assessment is instead based on establishing a long-term context for identifying exceptional severity and impact through convergent paleoclimatic proxies.

Our methodology for the Tang period relies on two primary lines of evidence that move beyond simple deviation from a mean:

Relative Severity Within Multi-Millennial Proxy Records:
We use high-resolution paleoclimatic archives that provide a context spanning millennia to identify events that are exceptional in their duration and magnitude.

- The tree-ring width chronology from Hessler et al. (2018), which reconstructs summer Palmer Drought Severity Index (PDSI) for central Mongolia (49 BCE–2011 CE), explicitly identifies the 19-year drought beginning in 804 CE as "one of the most extreme events in their entire record." This is a direct assessment of extremity based on its rarity and severity over a 2000-year period.
- The 3,476-year precipitation reconstruction for the northeastern Tibetan Plateau (Liu et al., 2025) indicates that the period encompassing the Sui and Tang dynasties (581–960 CE) was a "prolonged phase of generally dry conditions" that was distinctly drier than the preceding centuries. Within this multi-millennial context, the late Tang represents a significant and sustained negative anomaly.

Regarding the 900–907=8 CE period: The reviewer is correct that a marked decline in runoff alone may lack a defined baseline. Our classification of the terminal Tang period is strengthened by its position within the longer arid phases identified by Hessler et al. (2018). Our attempt is to provide the primary paleoclimatic evidence for *defining* the drought, but to discuss the documented *societal consequences* (crop failures, famine) of these climatically anomalous conditions in the historical record, thereby illustrating their "extreme" impact.

Spatial Heterogeneity: The apparent contradiction with the Huangye Cave speleothem record (Tan et al., 2011) is acknowledged in our manuscript as evidence of significant spatial heterogeneity. This does not invalidate the occurrence of extreme drought in the YRL region but clarifies that the extremes were regional. The documented societal crises are attributed to these severe regional anomalies.

In summary, our manuscript defines "extreme" events based on:

- Their exceptional duration and severity within multi-millennial proxy records.
- Their transformative environmental impact, as evidenced by geomorphological changes like desertification.

POINT 5

We agree that documentary sources can be a valuable source of knowledge, however, this is beyond the scope of this article. We cited secondary literature that is based on these data and the reader can refer to the papers to harvest the required information.

POINT 6

Thank you for this comment. Certainly, we considered to enlarge the temporal depths of this article to different periods, however, a cross-chronological analysis would clearly be beyond the scope of this paper. It would be a different research question, hypothesis and using different data. It would be a different paper or rather a book. We provide the code and the model framework so this can be done by follow-up projects and research also from other scholars. But even though it is not feasible to solve this on top of the present study, we added a critical nuance to our conclusion to stress this aspect and encourage future research to apply this model in other periods and test the resilience of the Tang Dynasty to contemporaneous ecological constraints compared to the resilience of other socio-political systems (e.g. in the previous or posterior periods)

Other issues:

1. The paleoclimate reconstruction sites used in the study should be clearly marked on the corresponding figures, like Fig 1.

- We cited the literature where the reader can find the location of the respective sample sites and areas.

2. In Figure 1, it would be helpful to include additional historical context by labeling the Tang dynasty capital, major administrative centers, key granaries (grain distribution hubs), and other relevant background information. These additions would enhance the reader's understanding of the geographic and political landscape referenced in the manuscript.

- We did not want to overload the map with additional information or background noise, hence we focused only on the relevant information that is important in this article

3.Fig.1: "annual average temperature (C) and precipitation (D) (Karger et al 2017)", the meaning of the "(>+15) and (<-15)" shown in panel C is unclear, the authors should explicitly explain what this threshold represents.

- These are not thresholds but the breaks for the colorbar and the data range.

4.Fig.2: the meaning of each panel should be clearly indicated in the figure caption. Additionally, the symbols or data points shown in the figure should be explained in the caption.

- Thank you for this. We edited the caption accordingly. The panels are zooms of the large map to facilitate readability.

Reviewer 2

Dear Reviewer #2,

Many thanks for your positive and constructive suggestions and for taking your time to review our manuscript! We are delighted that our model meets your scientific and interdisciplinary ideas and we would like to answer your suggestions point-by-point in the following section.

In Fig. 1: I suggest to flip the elevation colour bar (low at the bottom).

- We have overseen that and flipped the colour bar accordingly!

Fig. 2: It is unclear why the drought vulnerability index has small values for high vulnerability and high values for low vulnerability. From a logical point of view, it would make more sense to have large values for high vulnerability.

- We understand your concern and understand this can be misleading. Drought indices, such as the scPDSI range from negative values to positive values, referring to high (negative) to low (positive) drought vulnerability. We adapted to that approach in the paper to keep to the standardized index range (see for example here: <https://climatedataguide.ucar.edu/climate-data/palmer-drought-severity-index-pdsi>)

Fig. 3: The colours are not easy to see. In particular, the green shaded area of the "Tang period cropland" is not visible. The colours of "Suitability" and "Crop transport intensity" are not easy to assign in the map as the colours are the same.

- We apologize for the inconvenience! Color codes and appearances can change during PDF conversion etc and we were not aware of that issue during submission. We have

now adapted the color codes to increase readability of the map: the transport intensity is now in color style “magma” and the land-use probability has been adapted to a more focused color range based on the variation of the data spectrum. The barplot has been adapted accordingly.

Fig. 4: In panel (C), there are two shaded areas. Meaning unclear.

- Thank you, we did not mention that! The blue shaded line is the uncertainty of the dataset that comes with the record and the greyish envelope is the regression smoothing estimate from LOESS (local estimates). We added the respective information to the caption.

In the caption: "... characterize below normal run-off (potential flooding) (blue) or above normal run-off ..." – below normal and above normal seems to be swapped.

- Thank you so much for spotting that detail!! That was clearly a mistake and happened during fine-tuning the figure. We have flipped the words accordingly!

Line 335pp: add references.

- We added two more references to the sentence/the paragraph to back up this connection:

Skaaf 2000 (<https://doi.org/10.1179/072924700791201658>)

Graff 2017 (doi:10.1017/jch.2016.35)

- We also added a large literature review to the paper to meet the expectations of historical background.

Fig. 5 (and related results discussed in the text): It is unclear what the different line widths of the links mean in the lower panel means. Perhaps there is no meaning? But then I wonder why the links are thicker. I also suggest to increase the line width of the links in the other two panels in general, as they are quite thin.

- The different line thickness in the lower panel is caused by the IDW interpolation (see below comment) that has been performed with points sampled from the weighted vulnerability lines. These cause the effect of “spatial overlay” which is why we chose a rather small size for the interpolation (spatial distance). Here, the different size of a line is actually not a line anymore but a raster of that line. The vulnerability has been interpolated as rasters because in a line, the vector information would be given as one value one (e.g. a line is “1”), whereas a raster can have a gradient, continuous value or variable value ranges. This is why the “lines” appear to be in different size. We added more information to the text to ensure the workflow: “In Fig. 5 (lower panel), the

vulnerability of travelling across the landscape is visualized as an interpolated raster based on points sampled from the simulated connections between the sites. At each point, vulnerability values reflect the combined influence of flood and drought conditions along the underlying route segments, and when interpolated these values spread outward and overlap, producing broad corridors where exposure is consistently high”

- We updated the caption and text accordingly. We tried several line thicknesses, however, if the lines get thicker, they visually merge with adjacent lines and hence can not be differentiated individually... I hope that this version and the high resolution images available help to identify each single connection.

Is there any evidence from local sites for these road networks? I suppose that it might be possible to find some artifacts or other remnants along the ancient roads?

- Unfortunately, we cannot verify this as it is out of the scope of this paper to screen all the archaeological reports (in Chinese). Future research could verify local road infrastructure, bridges, or the distribution of artefacts. Due to the “nature” of ancient roads, we expect them, however, to be rarely “constructed” and most material will be gone by now.

Abbreviation IDW should be explained (is only explained at the end of the manuscript in the methods section).

- Yes, we clarified in the caption prior to the methods.

Lines 475pp: Is the suitability equation some standard approach? I wonder why it is a simple linear combination. Any evidence for this? And how to select the coefficients (aside of coefficient c). And how was it ensured to have a value range between 0 and 1? From the equation it is not clear (and could have larger values, I guess).

- The equation is a custom linear approach, that is correct. We have tried several ways to weigh the input parameters and in different settings, environmental conditions etc, these factors can be adapted accordingly. It is subjective as we decided to keep the model variables to a minimum, however, ensuring enough complexity to get a comprehensive model. The value range derives from the normalization process prior to fitting the model, as shown in the methods section.

Line 562: I suggest to use italic font only for the variables. The italic style for the sign of absolute values looks more like a sign for division. At first I was confused.

- Absolutely, that got overseen in the process, many thanks!!

Line 902: I suppose the figure references are not correct and should be Figs. S15 and S16, where they are showing not administrative and military sites in the different figures but the connectivity (S15) and permeability (S16).

- Right! Thank you for spotting this. We changed the description accordingly.

Fig. S16: Should there be differences in the lines shown between the upper and the lower panel? I cannot see any difference (only admin and military sites aer different).

- No, there are no differences in the density estimate of the connectivity, it is the same. The background, the lines, are a representation of the overall connectivity of the study area and serve as background information to the site distribution of military and administrative sites. Both distributions refer to the same overall connectivity. We decided for two panels to increase comparison and readability for each site pattern. The density estimate, which underlies the black lines (actually a raster), is calculated from regular points all across the study area. Hence, it is temporally independent.

Reviewer 3

Dear Reviewer #3,

We thank you for your positive evaluation of the manuscript! We agree that the development of multicomponent models now has entered a stage of complexity that enables us to draw multiple conclusions for past resource exploitation and land-use strategies, and even future forecasting and predictions. We are particularly delighted that you value the model and the reproducible research approach we are following. As you can see from the track-changes version, the article has undergone a major revision to meet all the criteria raised during the review process. We hope that this fine-tuned version is now even better contributing to the overall interdisciplinary approach. Thank you again for your time and work!